# DeepEMD: A Transformer-based Fast Estimation of the Earth Mover's Distance

## Abstract

The Earth Mover's Distance (EMD) is the measure of choice between point clouds. However the computational cost to compute it makes it prohibitive as a training loss, and the standard approach is to use a surrogate such as the Chamfer distance. We propose an attention-based model to compute an accurate approximation of the EMD that can be used as a training loss for generative models. To get the necessary accurate estimation of the gradients we train our model to explicitly compute the matching between point clouds instead of EMD itself. We cast this new objective as the estimation of an attention matrix that approximates the ground truth matching matrix. Experiments show that this model provides an accurate estimate of the EMD and its gradient with a wall clock speed-up of more than two orders of magnitude with respect to the exact Hungarian matching algorithm and one order of magnitude with respect to the standard approximate Sinkhorn algorithm, allowing in particular to train a point cloud VAE with the EMD itself. Extensive evaluation show the remarkable behaviour of this model when operating out-of-distribution, a key requirement for a distance surrogate. Finally, the model generalizes very well to point clouds during inference several times larger than during training.

## 1 Introduction

The *earth mover's distance* (EMD), also known as *Wasserstein distance* is a distance between distributions that is defined as the minimum total of mass-time-distance displacement needed to transform one distribution to the other. In the case of uniform distributions over a finite number of points, it turns into a distance between point clouds that corresponds to finding the one-to-one matching that minimizes the sum of the distances between pairs of matched points. Since there is no inherent ordering in point cloud data, computing the EMD between two point clouds involves finding a matching based on the euclidean distance between points. The matching is constrained to be bipartite so that one point cloud is completely transformed to the other, without any fractional assignment, and the transport cost is minimal.

The EMD is the most commonly used distance metric on point clouds, and is extremely useful in many different contexts. In particular as we will see for both assessing the performance of, and for training variational autoencoders, since the generated point cloud should get as close as possible to the target in terms of displacement. It can also be interpreted as the distance between two distributions computed with a finite number of samples and reflects the notion of nearness properly, does not have quantization/binning and non-overlapping support problems of most other metrics, e.g., $f$-divergences, total variation distance, etc.

The EMD between point clouds can be computed exactly, but it is extremely expensive computationally. The standard method is the Hungarian matching Kuhn (1955) algorithm whose complexity is $O(N^3)$ where $N$ is the number of points. Due to this computational cost, training deep generative models for point clouds is not done with this metric, even though it is a metric of choice for performance evaluation. The standard approach uses the *Chamfer distance* (CD) as surrogate. This metric can be computed in $O(N^2)$ time complexity but relaxes the one-to-one matching, which may create pathological situations.

We propose a deep architecture that takes as input two point clouds encoded as sets of geometric coordinate tuples, and computes an accurate estimate of the EMD. We show that the most efficient

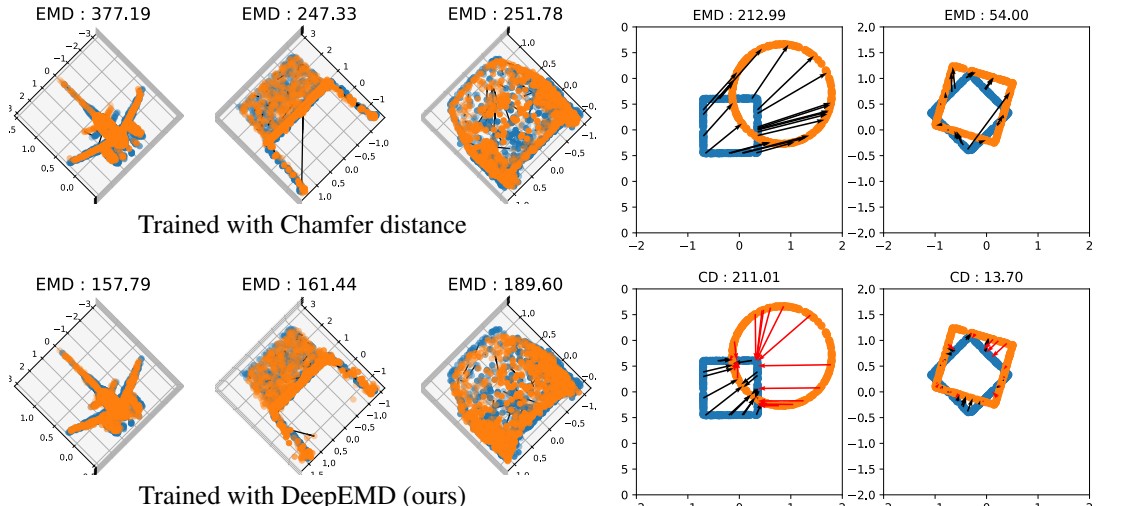

Figure 1: Example point clouds (blue) and their VAE reconstructions (orange) when trained with different reconstruction losses. Training with DeepEMD (bottom) consistently achieves lower reconstruction error (EMD, shown on top of each example) than with the standard Chamfer distance (top).

Figure 2: Example pairs of point clouds. The true earth mover's distance (EMD, top) and Chamfer distance (CD, bottom) are shown above each example. Arrows indicate the matching between the two point clouds under their respective metrics.

approach, in particular if that estimate is used as a loss for a training process, is to estimate the matching matrix itself. Since the EMD is the sum of the distance between matched points, this approach provides a very accurate estimate of the gradient with respect to the point coordinates. Training a deep variational autoencoder with our model instead of an exact computation is up to $\times 100$ faster (see Fig. 8), and the resulting model performs far better than one trained with the usual Chamfer surrogate (see § 4.4 and Fig. 7).

The key contributions of this paper can be summarized as:

- We propose DeepEMD which approximates the EMD between point clouds in $O(N^2)$ time complexity vs $O(N^3)$ of the hungarian algorithm (§ 3).

- We propose to cast the prediction of a bipartite matching as an attention matrix from which we get an accurate estimate of the EMD and its gradient (§ 3.2).

- We show that DeepEMD generalizes well to unseen data distributions (§ 4.3), and can be used for evaluation of generative models. It provides accurate estimates of the gradients of the distance and demonstrate that DeepEMD can be used as a surrogate reconstruction loss for training deep generative models of point clouds (§ 4.4).

- We show that DeepEMD achieves about $40\times$ speed-up over Sinkhorn algorithm, achieving equal or better performance for various metrics (§ 4.3).

## 2 RELATED WORK

The two commonly used distance metrics for point clouds in literature are Earth Mover's Distance (EMD) and Chamfer Distance (CD). Consider two point clouds $X = \{x_i\}_{i=1}^N$ and $Y = \{y_j\}_{j=1}^N$, where $x_i, y_j \in \mathbb{R}^d$. The EMD between the two point clouds can be computed as,

$$\text{EMD}(X, Y) = \min_{\phi \in \mathcal{M}(X,Y)} \sum_{x \in X} \|x - \phi(x)\|_2, \tag{1}$$

where $\mathcal{M}(X, Y)$ is the set of 1-to-1 (bipartite) mappings from $X$ to $Y$. In addition to the distance, the optimal matching $\phi^*$ is also interesting for some applications. Since directly optimizing the EMD is computationally expensive, most methods in the literature rely on CD as a proxy similarity measure or reconstruction loss. The CD can be computed as,

$$\mathrm{CD}(X, Y) = \sum_{x \in X} \min_{y \in Y} \|x - y\|_2^2 + \sum_{y \in Y} \min_{x \in X} \|x - y\|_2^2, \tag{2}$$

and in $O(N^2)$ time complexity. The CD solution leads to a non-bipartite one-to-many matching between $x \rightarrow y$ and vice versa. We can also use the $L_2$ measure with $d = \|x - y\|_2$ instead $d = \|x - y\|_2^2$ to make it comparable to EMD. of Note that the above EMD for point clouds is related to the Wasserstein-2 metric (see appendix § A for details). The utility of EMD is limited by the $O(N^3)$ computational cost of evaluating it. There have been several research efforts to circumvent this issue in various application settings.

This is the case for application to point clouds where $N$ is usually in the range of several thousands. Kim et al. (2021) trains a variational auto-encoder with CD as the reconstruction loss. EMD is still the metric of choice for evaluating point cloud generative models Huang et al. (2022); Luo and Hu (2021); Kim et al. (2021); Yang et al. (2019); Shu et al. (2019); Achlioptas et al. (2018). Another issue is disparity between performance measures (minimum matching distance, coverage, etc.) computed with EMD and CD, the comparisons are contradictory and often inconsistent across measures.

CD is usually insensitive to mismatched local density while EMD is dominated by global distribution and overlooks the fidelity of detailed structures Wu et al. (2021). Wu et al. (2021) proposes a new similarity metric called Density-aware Chamfer distance (DCD) to tackle these issues. DCD is derived from CD and can also be computed in $O(N^2)$ time complexity. Urbach et al. (2020) proposed Deep Point Cloud Distance (DPDist) which measures the distance between the points in one cloud and the estimated continuous surface from which the other point cloud is sampled. The surface is estimated locally by a network using the 3D modified Fisher vector representation.

Shirdhonkar and Jacobs (2008) proposed a linear time algorithm for approximating the EMD by exploiting the Hölder continuity constraint in its dual form to convert it into a simple optimization problem with an explicit solution in the wavelet domain and computed as the sum of absolute values of the weighted wavelet co-efficients of the difference histogram. However, their approach is limited to low dimensional histograms. In the optimal transport literature, several efforts have been taken towards improving the statistical and computational properties. Recently, Chuang et al. (2022) proposed Information Maximizing Optimal Transport (InfoOT) which is an information-theoretic extension of optimal transport based on kernel density estimation of the mutual information which introduces global structure into OT maps. The resulting solution maximizes the mutual information between domains while minimizing geometric distance and improves the capability for handling data clusters and outliers.

Other approaches focus on regularizing the OT problem for making it smooth and strictly convex Cuturi (2013); Flamary et al. (2016); Genevay et al. (2018); Blondel et al. (2018). *Sinkhorn distances* Cuturi (2013) smooth the classic OT problem with an entropic regularization term and can be computed through Sinkhorn's matrix scaling algorithm at a speed that is several orders of magnitude faster than that of transport solvers. We provide more details in the appendix. Meta OT Amos et al. (2022) proposes a meta model to predict the solution to the optimal transport problem which is then used to initialize a standard Sinkhorn solver to further refine the predicted solution. The architecture of the meta model depends on the data domain, and DeepEMD can be utilized when working with point clouds. The choice of meta model architecture is contingent upon the specific data domain, and DeepEMD demonstrates its exceptional utility in point cloud processing.

In this paper, our goal is to approximate the EMD using a deep network in a learning based paradigm where each sample represents two distributions and the target for regression is either the true metric i.e. EMD or the optimal matching $\phi^*$, or both. Existing point cloud datasets can serve as an interesting learning problem, where we can interpret each point cloud as a 2D or 3D distribution of points on a shape (manifold). It is posed as a supervised learning problem where the task is to estimate the true EMD, or the true bipartite matching, or both, between a pair of input point clouds.

## 3 METHOD

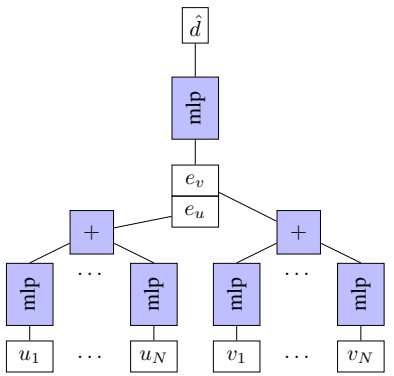

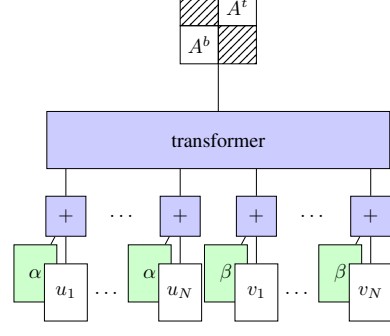

Figure 3: The MLP model (see § 3.1) predicts directly an estimate $\hat{d}$ of the EMD.

Figure 4: The transformer model we use for Deep-EMD (see §3.2) predicts directly the bipartite graph as an attention matrix.

We are interested in building a model which operates on a pair of point clouds $(U, V)$ as input, where $U, V \in \mathcal{D}^N$, $U = \{u_i\}_{i=1}^N$, $V = \{v_j\}_{j=1}^N$, $u_i, v_j \in \mathbb{R}^D$, and $N$ is the cardinality of the point clouds. Note that the points are unordered and the indexing is arbitrary. We denote the earth mover's distance between them as $d = \text{EMD}(U, V)$ where $d \in \mathbb{R}$. The goal of the model is to predict $d$ and $\nabla d$. Also, let $M \in \{0, 1\}^{N \times N}$ denote the ground truth bipartite matching from EMD, where $M_{i,j} = 1$ indicates that $u_i$ is matched to $v_j$ and vice-versa. Bipartite-ness implies $\forall j, \sum_i M_{i,j} = 1$ and $\forall i, \sum_j M_{i,j} = 1$.

Since point clouds are unordered and invariant to elementwise permutation, we seek mappings $f : \mathcal{D}^N \times \mathcal{D}^N \to \mathbb{R}$ which are permutation invariant for any permutations $\pi$ and $\pi'$, i.e.,

$$f(U, V) = f(\pi(U), \pi'(V)), \tag{3}$$

In the following sections, we first introduce a simple MLP based baseline, followed by our transformer-based model, *DeepEMD*.

### 3.1 PREDICTING THE DISTANCE

We propose a simple MLP baseline composed of a point-wise MLP backbone, followed by a prediction head which is also a MLP (see figure 3). The backbone MLP takes a point cloud and returns an embedding $e \in \mathbb{R}^d$ as,

$$e_u = \sum_{i=1}^n g(u_i), \quad e_v = \sum_{j=1}^n g(v_i) \tag{4}$$

The prediction head then produces the final prediction as,

$$\hat{d} = h\left(e_u \oplus e_v\right) + h\left(e_v \oplus e_u\right), \tag{5}$$

where $\oplus$ denotes vector concatenation. Both $g$ and $h$ are composed of sequential linear layers with ReLU non-linearity between layers. The embeddings are permutation equivariant because of the sum aggregation which does not depend on the ordering of points. Further, we concatenate the embeddings both ways as in Eq. (5), which makes the mapping symmetric. We train the model with mean-squared error loss, $l = (d - \hat{d})^2$. Since the model does not predict the matching, we can interpret it from the direction of the gradient of a point $\delta v_j = \left[\frac{\partial \hat{d}}{\partial V}\right]_j$, e.g., by taking cosine similarity between $\delta v_j$ and $u_i - v_j$, where $u_i$ is the point matching to $v_j$ from EMD.

### 3.2 PREDICTING BIPARTITE MATCHING

The transformer Vaswani et al. (2017) intuitively seems to be a very good model for reasoning with point clouds and matching. Moreover, by considering that the output is the last layer's attention

matrix, we can use it to directly predict the bipartite matching. Since the EMD–and consequently its gradient with respect to the point positions–is a function of the point positions and the matching array, predicting the latter leads to an accurate estimate of the former, that in particular is shielded from the issue of a possible decoupling between matching a functional point-wise (e.g. for MSE) and matching the gradients. While it is viable to directly predict the distance with a transformer-like architecture, we chose to only predict the matching since it is straightforward to estimate the distance using the predicted matching.

We propose *DeepEMD* composed of a sequence of multi-head full attention layers, followed by a prediction head which is also a full attention layer, but with a single head. Given two point clouds $U$ and $V$, add a learned cloud-specific positional embedding to indicate if a point originates from $U$ or $V$, we concatenate the points sequence and feed the resulting $I = U \cup V$ as input to the model. The group-id embedding helps the model in modulating attention locally within a point cloud as well as globally across both point clouds. We tried other variants with self-attention layers, cross attention layers, and an alternating mixture of both, but found full attention over both point clouds to be best performing.

For our problem, we get the input $X$ for the transformer by adding positional embeddings to $I$. Let $\vec{0}^n$ and $\vec{1}^n$ denote a vector of $n$ zeros and ones, respectively. $X$ is then obtained as,

$$P = \vec{0}^n \cup \vec{1}^n, \quad X = I + W^P[P] \quad \text{(indexing)} \tag{6}$$

The intermediate feature $t(X)$ from the transformer encoder (see appendix for details) has the same number of elements as $X$ with each element now being a contextualized representation for the corresponding point in the input. Further, these intermediate representation are fed into a single-head attention layer which outputs the attention matrix as,

$$K = t(X)W^K, \quad Q = t(X)W^Q, \quad A = \frac{QK^\top}{d_k} \tag{7}$$

$$A^t = A_{:n,n:} \quad A^b = A_{n:,:n} \tag{8}$$

Here, $A$ is a $2N \times 2N$ matrix and we slice the top-right block (first $N$ rows and last $N$ columns) as $A^t$ and bottom-left block (last $N$ rows and first $N$ columns) as $A^b$. $A^t_{i,j}$ can be interpreted as the relatedness of $u_i$ with $v_j$. Similarly, $A^b_{i,j}$ can be interpreted as the relatedness of $v_i$ with $u_j$.

Given $M$, the ground truth bipartite matching from EMD, we define the loss as the average of the cross-entropies as,

$$l(U, V) = \frac{1}{N} \sum_{i=1}^{N} \text{CE}(A^t_{i,.}, M_{i,.}) + \frac{1}{N} \sum_{i=1}^{N} \text{CE}(A^b_{i,.}, M_{.,i}) \tag{9}$$

The EMD is then estimated with the predicted matching as,

$$\phi^b(i) = \underset{j}{\text{argmax}}\, A^b_{i,j}, \quad \phi^t(i) = \underset{j}{\text{argmax}}\, A^t_{i,j} \tag{10}$$

$$\hat{d} = \frac{1}{2} \left( \sum_i \|u_i - v_{\phi^t(i)}\| + \sum_i \|v_i - u_{\phi^b(i)}\| \right) \tag{11}$$

## 4 EXPERIMENTS

In this section, we present the overall experimental setup, performance results and comparisons of DeepEMD across various tasks.

### 4.1 DATASETS

We consider different datasets for our experiments - *Syn2D*, ShapeNet Chang et al. (2015), ModelNet40 Wu et al. (2015) and ScanObjectNN Uy et al. (2019). *Syn2D* consists of 2D point clouds generated synthetically by sampling points on squares and circles (see Fig 2). ShapeNet and ModelNet40 are datasets of 3D point clouds derived from 3D CAD models for different real world objects

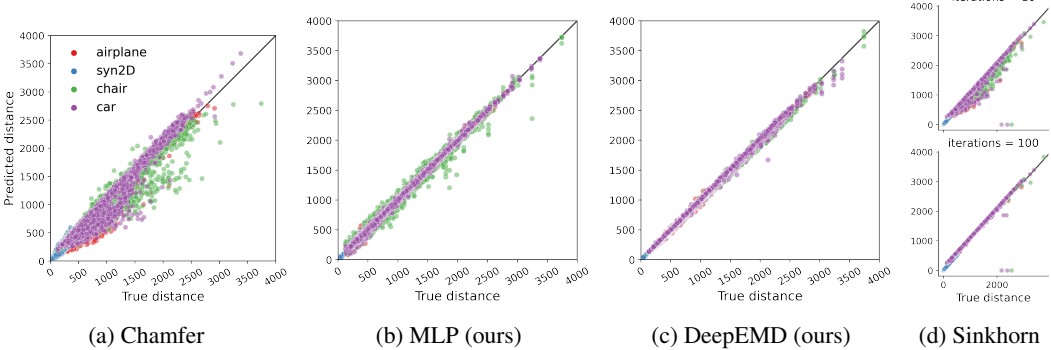

(a) Chamfer      (b) MLP (ours)      (c) DeepEMD (ours)      (d) Sinkhorn

Figure 5: Scatter plot for true vs. approximate EMD from different models/metrics on validation splits for Syn2D and ShapeNet datasets. DeepEMD (ours) consistently performs better across different categories as it has less dispersion. Sinkhorn algorithm becomes more accurate with more iterations. Also note that it encounters numerical errors for some examples.

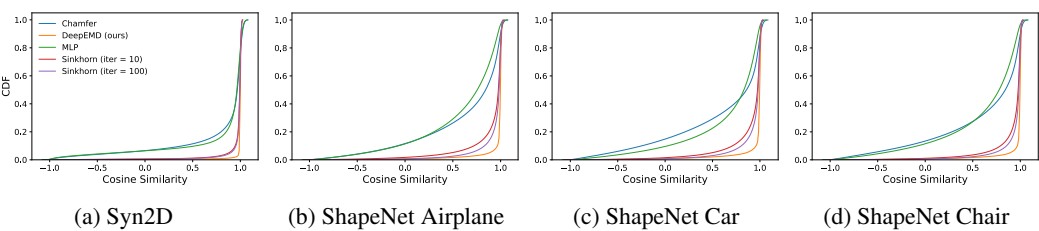

(a) Syn2D      (b) ShapeNet Airplane      (c) ShapeNet Car      (d) ShapeNet Chair

Figure 6: CDF of cosine similarity between true and estimated gradients for all points across all point clouds collected together on validation splits for Syn2D and ShapeNet datasets. The ideal cdf curve should have all the mass at cosine similarity 1. DeepEMD (ours) consistently outperforms all the other methods across different datasets.

like chairs, cars, airplanes, etc. ScanObjectNN is a relatively new real-world point cloud object dataset based on scanned indoor scene data. In order to improve and assess generalization, we augment the train and test splits with synthetic perturbations. We provide more details about the datasets and these augmentations in the Appendix § B.

## 4.2 PERFORMANCE MEASURES

We consider various measures to assess EMD approximation methods for distance as well as matching estimation. We compare accuracy and computation time to that of Sinkhorn and CD (see § 2).

**Distance Estimation.** We visualize the true vs. predicted distance through scatter plots (Fig. 5), we expect the data points to be close to $x = y$ line. We compare various correlation measures : linear correlation ($r$), Spearman correlation ($\rho$) and Kendall-Tau correlation ($\tau$), to assess the quality of distance estimation. The Spearman and Kendall-Tau are rank-statistic based correlation measures, indicative of the correspondence between two rankings. Note that, correlation measures are useful metrics as they indicate appropriateness of the predicted metric as a distance measure, irrespective of their absolute values. Additionally we look at different quantiles ($\text{RE}_n$) of relative approximation error, which penalizes the difference between absolute values of the predicted and true distance.

**Matching Estimation.** In order to assess quality of the matching, we consider the cosine similarity between the true and predicted gradient. The true gradient of EMD is always along the matched point. We visualize the cumulative distribution function (cdf) of cosine similarities (Fig. 6), where we expect all the mass to be close to 1. We also look at different quantiles ($\text{CS}_n$) of the cosine similarity. We also consider accuracy which is computed as the average accuracy of matching source points to target points and vice-versa, bipartiteness (B) which is fraction of points with bipartite

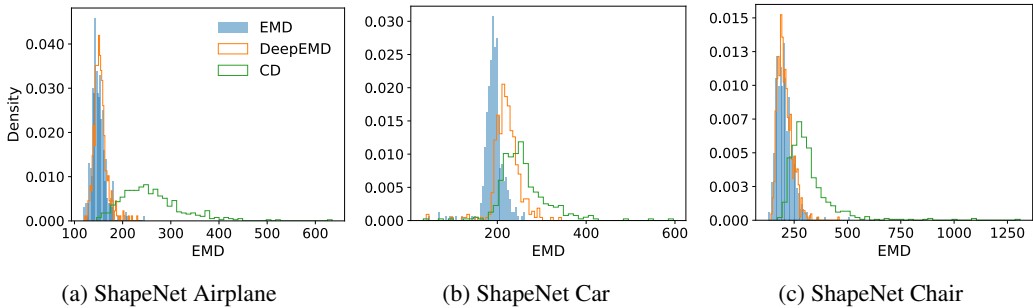

(a) ShapeNet Airplane      (b) ShapeNet Car      (c) ShapeNet Chair

Figure 7: Comparison of EMD between input and reconstructed point clouds from SetVAE trained with different reconstruction losses. The better model should have smaller reconstruction loss and thus mass close to zero in the histograms. DeepEMD (ours) is consistently better as compared to Chamfer loss and very similar to EMD loss.

matching, and also bipartiteness-correctness ($B_{corr}$) which is fraction of points which are bipartite as well as matched correctly.

## 4.3 RESULTS

**EMD Prediction.** Fig. 5 shows the scatter plot of the true EMD vs. approximate EMD predicted from our trained models on the validation split for *Syn2D* and ShapeNet datasets. Note that the validation split also contains the augmentations as discussed in Sec. 4.1. We also validate on specific splits for which the results are shown in the appendix. The plots indicate that both DeepEMD (Fig. 5c) and MLP baseline (Fig. 5b) approximate the EMD faithfully. The MLP baseline seems to struggle a bit on ShapeNet Chair dataset. The higher dispersion in Chamfer (Fig. 5a) and Sinkhorn with 10 iterations (Fig. 5d, top) indicates poor EMD estimation. The approximation with Sinkhorn algorithm becomes more accurate with higher number of iterations (Fig. 5d, bottom), as expected.

We summarize various metrics in Tables 5, 6 and 7 the appendix. DeepEMD and MLP baseline both achieve linear correlation higher than 0.99 in each case. The models achieve Kendall-Tau correlation close to 0.99 and 0.96, respectively, and Spearman correlation close to 0.99 in each case, indicating that ordering of the samples based on approximate distances and true distances are very similar, and monotonocity of samples are preserved. It can be observed that DeepEMD is best except for relative error where it may do worse than our method with MLP. Also it is interesting to note that Sinkhorn performs worse than DeepEMD on correlation- or relative error-based measures.

**Matching/Gradient Prediction.** Estimating the matching and gradient of the distance is particularly important for training models with DeepEMD as a surrogate distance function. Note that the gradient of a point from true EMD is always along the matched point in the other point cloud. Fig. 6a shows the cdf of cosine similarity between the true and estimated gradient for all the points across all point clouds collected together for *Syn2D*, while Figs. 6b, 6c, and 6d for ShapeNet. The cdf has most mass at cosine similarity close to 1 with a very short tail and is never negative indicating that the estimated gradient is aligned with the true gradient for DeepEMD. This is particularly important when the model is used as a surrogate reconstruction loss. Ideally, the model should provide good estimate of the true gradient throughout training and more particularly in the very beginning when the reconstructions are very noisy, and also towards the end when reconstructions likely become very similar to the training distribution. We discuss more on this in the next sections and the appendix. The MLP baseline usually did not perform well in this regard and also in generalizing to unseen examples and thus was not useful for training generative models. The same issue can be observed with Chamfer distance as well. Table 5 (appendix) shows the cosine similarity quantiles ($CS_n$), as well as accuracy, bipartiteness (B) and bipartiteness-correctness ($B_{corr}$). DeepEMD performs better than other models and metrics in each of these measures and indicates better matching approximation.

**Out-of-distribution generalization.** The genralization of the prediction to a novel distribution is particularly importatnt for a surrogate metric. We test the out-of-distribution behaviour of our models in two different settings : Table 1 shows the generalization performance of the model trained on

a single category of ShapeNet and tested on validation split of multi-category ModelNet40 dataset, while Tables 8, 9 and 10 in the appendix show the performance when tested on different ShapeNet categories. The results indicate that DeepEMD generalizes well when test and train data differ without any adaptation or fine-tuning. Further, the validation performance on a category of a model trained on another category (see Appendix for details) is very similar to the performance of the model trained on the same category. These quite remarkable behaviors point towards the network "meta-learning" in some way the matching algorithm. This is further strengthened by the results on scaling to different number of points during test time as shown in Table 2.

Table 1: Out-of-distribution (dataset) generalization for our models and comparison with other metrics (Chamfer and Sinkhorn), tested on full validation split for ModelNet40 (with 40 categories) and ScanObjectNN (with 15 categories). The models are trained on a single ShapeNet category. The reported numbers are averaged over these categories as well as four training seeds. The first six rows show distance estimation metrics (see § 4.2), while the last six rows correspond to matching estimation metrics. The arrows next to the metrics indicate whether higher ($\uparrow$) values are better or lower ($\downarrow$). Chamfer and Sinkhorn are deterministic, thus variances are not reported. Further, MLP does not provide accuracy and bipartiteness metrics.

| DATASET | MODELNET40 | | | | SCANOBJECTNN | | | |
|---|---|---|---|---|---|---|---|---|
| MODEL | CHAMFER | SINKHORN | MLP | DEEPEMD | CHAMFER | SINKHORN | MLP | DEEPEMD |
| $r$ ($\uparrow$) | 0.951 | 0.971 | $0.959 \pm 0.011$ | $\mathbf{0.999} \pm 0.0$ | 0.971 | 0.929 | $0.965 \pm 0.005$ | $\mathbf{0.997} \pm 0.001$ |
| $\rho$ ($\uparrow$) | 0.935 | 0.988 | $0.945 \pm 0.017$ | $\mathbf{0.999} \pm 0.0$ | 0.979 | 0.965 | $0.963 \pm 0.007$ | $\mathbf{0.999} \pm 0.0$ |
| $\tau$ ($\uparrow$) | 0.792 | $\mathbf{0.983}$ | $0.819 \pm 0.024$ | $0.974 \pm 0.002$ | 0.882 | 0.968 | $0.855 \pm 0.011$ | $\mathbf{0.973} \pm 0.002$ |
| $RE_{0.1}$ ($\downarrow$) | 0.03 | 0.057 | $0.009 \pm 0.001$ | $\mathbf{0.005} \pm 0.002$ | 0.025 | 0.038 | $0.013 \pm 0.001$ | $\mathbf{0.004} \pm 0.001$ |
| $RE_{0.5}$ ($\downarrow$) | 0.129 | 0.102 | $0.062 \pm 0.005$ | $\mathbf{0.019} \pm 0.004$ | 0.094 | 0.078 | $0.076 \pm 0.005$ | $\mathbf{0.019} \pm 0.004$ |
| $RE_{0.9}$ ($\downarrow$) | 0.321 | 0.2 | $0.257 \pm 0.03$ | $\mathbf{0.04} \pm 0.004$ | 0.282 | 0.244 | $0.299 \pm 0.025$ | $\mathbf{0.051} \pm 0.005$ |
| $CS_{0.1}$ ($\uparrow$) | $-0.067$ | 0.824 | $-0.293 \pm 0.047$ | $\mathbf{0.927} \pm 0.003$ | 0.138 | 0.879 | $-0.208 \pm 0.042$ | $\mathbf{0.946} \pm 0.002$ |
| $CS_{0.5}$ ($\uparrow$) | 0.834 | 0.986 | $0.684 \pm 0.023$ | $\mathbf{1.0} \pm 0.0$ | 0.917 | 0.992 | $0.719 \pm 0.02$ | $\mathbf{0.999} \pm 0.0$ |
| $CS_{0.9}$ ($\uparrow$) | 0.997 | 0.999 | $0.96 \pm 0.003$ | $\mathbf{1.0} \pm 0.0$ | 0.998 | 1.0 | $0.965 \pm 0.003$ | $\mathbf{1.0} \pm 0.0$ |
| ACCURACY ($\uparrow$) | 12.651 | 31.91 | - | $\mathbf{56.38} \pm 0.604$ | 7.673 | 20.04 | - | $\mathbf{40.671} \pm 0.62$ |
| B ($\uparrow$) | 17.045 | 33.458 | - | $\mathbf{70.401} \pm 0.672$ | 9.474 | 19.43 | - | $\mathbf{56.269} \pm 0.71$ |
| $B_{corr}$ ($\uparrow$) | 6.544 | 19.615 | - | $\mathbf{47.084} \pm 0.741$ | 3.38 | 9.961 | - | $\mathbf{30.055} \pm 0.678$ |

**Scaling number of points.** Remarkably, the size of point clouds during testing can differ greatly from those during training without degrading performance. Table 2 shows performance of the model for test point cloud sizes ranging from 256 to 8196, while training was done with only 1024 points. Prediction of the metric itself (top 6 rows) does not degrade for all practical purposes. Regarding the matching estimation, directional measure of performance related to the cosine similarity (rows $CS_n$) do not degrade neither. We can notice degradation in accuracy based measures (last 3 rows) which is natural since the problem becomes difficult with increasing number of points $N$ because of its combinatorial nature. For training when memory requirement is much higher due to backprop, we can use smaller number of points, and scale it up during inference without any fine-tuning.

Table 2: Scaling number of points and out-of-distribution (scale) generalization for DeepEMD. The models are trained on a single ShapeNet category with 1024 points and tested on validation split of same category but with different number of points. Reported values are averaged over 4 training seeds. DeepEMD generalizes well to unseen number of points at test time without fine-tuning.

| # points | $\longleftarrow$ Less # points than training $\longrightarrow$ | | | Trained | $\longleftarrow$ More # points than training $\longrightarrow$ | | |
|---|---|---|---|---|---|---|---|
| | 256 | 512 | 768 | 1024 | 2048 | 4096 | 8192 |
| $r$ | $1.0 \pm 0.0$ | $1.0 \pm 0.0$ | $1.0 \pm 0.0$ | $1.0 \pm 0.0$ | $1.0 \pm 0.0$ | $0.999 \pm 0.0$ | $0.999 \pm 0.001$ |
| $\rho$ | $1.0 \pm 0.0$ | $1.0 \pm 0.0$ | $1.0 \pm 0.0$ | $1.0 \pm 0.0$ | $1.0 \pm 0.0$ | $0.999 \pm 0.0$ | $0.998 \pm 0.0$ |
| $\tau$ | $0.985 \pm 0.0$ | $0.987 \pm 0.0$ | $0.988 \pm 0.0$ | $0.988 \pm 0.001$ | $0.986 \pm 0.001$ | $0.981 \pm 0.002$ | $0.974 \pm 0.004$ |
| $RE_{0.1}$ | $0.002 \pm 0.001$ | $0.002 \pm 0.001$ | $0.004 \pm 0.002$ | $0.007 \pm 0.003$ | $0.012 \pm 0.005$ | $0.013 \pm 0.007$ | $0.014 \pm 0.008$ |
| $RE_{0.5}$ | $0.01 \pm 0.002$ | $0.011 \pm 0.002$ | $0.014 \pm 0.003$ | $0.017 \pm 0.005$ | $0.027 \pm 0.009$ | $0.034 \pm 0.013$ | $0.04 \pm 0.016$ |
| $RE_{0.9}$ | $0.026 \pm 0.003$ | $0.026 \pm 0.003$ | $0.029 \pm 0.004$ | $0.032 \pm 0.005$ | $0.042 \pm 0.009$ | $0.054 \pm 0.013$ | $0.066 \pm 0.018$ |
| $CS_{0.1}$ | $0.94 \pm 0.002$ | $0.955 \pm 0.002$ | $0.961 \pm 0.001$ | $0.964 \pm 0.001$ | $0.967 \pm 0.001$ | $0.967 \pm 0.001$ | - |
| $CS_{0.5}$ | $1.0 \pm 0.0$ | $1.0 \pm 0.0$ | $1.0 \pm 0.0$ | $1.0 \pm 0.0$ | $1.0 \pm 0.0$ | $1.0 \pm 0.0$ | - |
| $CS_{0.9}$ | $1.0 \pm 0.0$ | $1.0 \pm 0.0$ | $1.0 \pm 0.0$ | $1.0 \pm 0.0$ | $1.0 \pm 0.0$ | $1.0 \pm 0.0$ | - |
| Accuracy | $72.348 \pm 0.44$ | $69.384 \pm 0.383$ | $66.901 \pm 0.379$ | $64.648 \pm 0.404$ | $57.588 \pm 0.464$ | $47.78 \pm 0.51$ | $35.274 \pm 0.483$ |
| B | $81.857 \pm 0.755$ | $80.101 \pm 0.547$ | $78.013 \pm 0.507$ | $75.896 \pm 0.521$ | $68.658 \pm 0.584$ | $58.109 \pm 0.597$ | $44.603 \pm 0.734$ |
| $B_{corr}$ | $64.838 \pm 0.756$ | $61.558 \pm 0.587$ | $58.545 \pm 0.547$ | $55.719 \pm 0.568$ | $46.831 \pm 0.618$ | $35.053 \pm 0.6$ | $21.606 \pm 0.469$ |

**Computational Time and Complexity.** Fig. 8 compares the evaluation time for different models and metrics. DeepEMD achieves a significant speedup of about $100\times$ as compared to EMD and $40\times$

as compared to Sinkhorn with 100 iterations. This speedup becomes more pronounced on bigger point clouds as hungarian algorithm takes $O(N^3)$ time vs. $O(N^2)$ for DeepEMD.

## 4.4 DEEPEMD USED AS A LOSS

Training a SetVAE, as for any auto-encoder, requires a reconstruction loss to assess the quality of the learned representation. While the eventual goal would be to minimize the EMD, standard approach uses Chamfer Distance due to the prohibitive computation cost of calculating the EMD. Instead of Chamfer Distance we propose to use DeepEMD and demonstrate its utility as a reconstruction loss as compared to Chamfer Distance.

DeepEMD was trained separately on each category of ShapeNet dataset and the trained model was then used as a surrogate reconstruction loss for training a variational auto-encoder. We use SetVAE Kim et al. (2021), a transformer based VAE adapted for point clouds and set-structured data. The parameters of DeepEMD module are frozen during training of the SetVAE. We follow exactly the same protocol as in SetVAE and train using ShapeNet categories of airplane, chair, and car and also the same hyper-parameters for training. Fig. 1 and Fig. 9 (appendix) shows the reconstruction on validation data achieved by SetVAE models trained with different reconstruction losses. DeepEMD consistently achieves lower reconstruction EMD as compared to CD. This is further verified from Fig. 7 which shows the distribution of true EMD between a point cloud and its reconstruction.

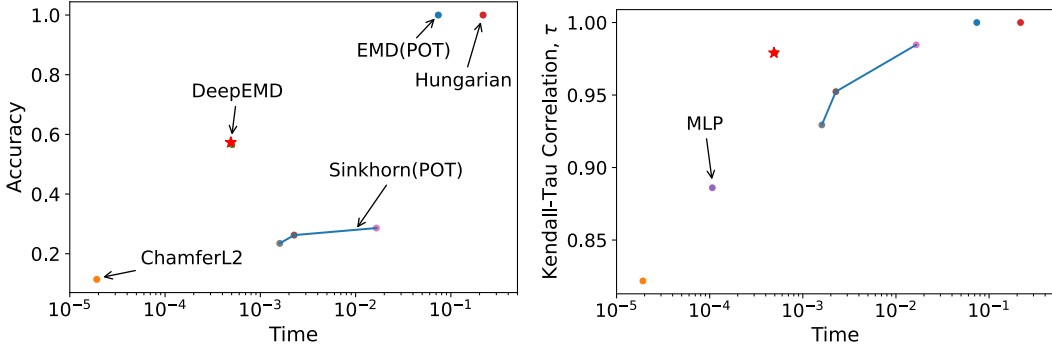

Figure 8: Comparison of empirical evaluation time and different performance measures : Accuracy (left) and Kendall-Tau correlation $\tau$ (right). We use Python Optimal Transport (POT) library for computing Sinkhorn distances, and show metrics at different iterations (5, 10 and 100). DeepEMD is $\sim 100\times$ and $\sim 40\times$ faster than Hungarian algorithm and Sinkhorn (100 iterations), respectively.

## 5 CONCLUSION AND FUTURE WORK

We propose DeepEMD, a method for fast approximation of EMD, improving time complexity from $O(N^3)$ to $O(N^2)$. It is composed of a multi-head multi-layer transformer, followed by a single-head full attention layer as the final output layer. It operates on two point clouds and outputs an attention matrix which is interpreted as the matching matrix and optimized to match the ground turh matching obtained from the hungarian algorithm. We demonstrated the effectiveness of DeepEMD in approximating the true EMD for synthetic 2D point clouds as well as real world datasets like ShapeNet, ModelNet40 and ScanObjectNN. It achieves a speed-up of $\times 100$ with 1024 points. Further, we show that it estimates the gradients well, generalizes well for unseen point clouds (or distributions), and can be used for end-to-end training of point cloud autoencoders achieving faster convergence than Chamfer distance surrogate.

It would be interesting to explore fast transformer variants to further improve from the quadratic time complexity for future work. In terms of architecture various pooling/un-pooling strategies can be explored which can help with both, better time complexity and improved feature learning. In this work, we estimate the Wasserstein $-2$ metric, and extension to other Wasserstein$-p$ metrics and other optimal transport problems could also be interesting for various applications. Lastly, extension to general probability distributions with fractional assignments (i.e. mass splitting) can also be very useful and valuable for some applications.

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
