# OpenReview forum: "DeepEMD: A Transformer-based Fast Estimation of the Earth Mover’s Distance"
_ICLR.cc/2024/Conference — Submitted to ICLR 2024_

### Official Review · Reviewer_yNgN · 2023-10-22

**Soundness:** 3 good
**Presentation:** 3 good
**Contribution:** 2 fair
**Rating:** 3
**Confidence:** 3

**Summary:**

This paper introduces an attention-based model designed to approximate the Earth Mover's Distance (EMD) for point clouds. Traditional EMD calculations are computationally expensive, so surrogates like the Chamfer distance are often used. To overcome this, the authors focus on computing the point cloud matchings rather than the EMD itself. Experiments show that this model can be used to estimate the EMD faster than using the Hungarian method or Sinkhorn algorithm. The paper demonstrates how the approach can be used to train a point cloud Variational Autoencoder (VAE) using the approximated EMD.

**Strengths:**

S1. Framing of the problem: The framing of the point-cloud matching problem as learning a cost function is quite intersting and makes me think of a number of applications for which this can be used. The empirical results provided, particularly the comparison against the Chamfer distance, the Hungarian method, and the Sinkhorn algorithm (as seen in Figure 8), is nice. This comparison gives a clear view of where the proposed model stands in terms of computational efficiency and accuracy.


S2. Clarity: The paper's structure and presentation seem to be well-organized, with the problem clearly articulated. The choice of casting the problem as an estimation of an attention matrix and its relevance is also clear. However, for a reader unfamiliar with point cloud analysis or attention mechanisms, some sections might be dense. A more thorough introduction or background on these concepts could enhance clarity.

S3. Significance: Notwithstanding missing comparisons (see Weaknesses below) I think the proposed model for approximating EMD could be quite useful in a number of scenarios (basically anywhere the Hungarian algorithm or the Chamfer distance is used at the moment). The model's performance, particularly in the out-of-distribution tests, is quite promising. The model also seems to generalize well to larger point-clouds than on which it was trained.

**Weaknesses:**

W1. Lack of Novelty: The problem of point-cloud registration is very well-studied, with numerous learning-based methods proposed over the years. Some of these also include attention mechanisms to do the matching. See the methods listed here: https://github.com/XuyangBai/awesome-point-cloud-registration

W2. Comparative Evaluation: The paper does not compare its method to state-of-the-art registration approaches. Given that point-cloud registration is a long-studied problem, there are likely several strong baseline methods against which the proposed model should be evaluated.

W3. Generalization may be an issue: While the paper mentions that the model generalizes well to larger point clouds during inference than those seen during training, most of the experiments are carried out on datasets consistent of synthetic 3D models. What happens when this is applied to real LiDAR scans or depth images?

W4. Scalability Issues: Attention mechanisms, particularly in the context of a large number of outputs, can be memory-intensive. Given that point clouds can be large (millions of points), it might be difficult to scale the proposed approach to such cases.

**Questions:**

I would suggest the following specific changes to address the weaknesses listed above:

W1. It would be helpful if the paper's contribution can be positioned more clearly within the existing landscape of point-cloud registration methods. Providing a thorough review of existing methods, especially those that leverage learning-based approaches, will help highlight the contributions of this work.

W2. The authors should compare their approach against leading methods in point-cloud registration, possibly ones from the above list. By doing so, they can better demonstrate the advantages of their method, be it in terms of accuracy, speed, or other criteria.

W4. The memory overhead of the attention-based method should be addressed, particularly in the context of large point clouds. Providing benchmarks for total memory usage would be beneficial.

---

> ### Author Response · Authors · 2023-11-16
>
> Thank you for your time reviewing our paper. We will try to address your concerns regarding related works, the weaknesses you highlighted, and your questions.
>
>
> W1,W2 : We are not solving a point cloud registration task. While we evaluate on 3D point clouds,
> DeepEMD does not assume anything about the input data and does not process 3D point clouds in
> any special way, in particular does not try to “match” according to a local appearance similarity.
> The main goal is to approximate EMD between two distributions which is ’similar’ to point cloud
> matching when working with point clouds as uniform distributions. In principle, DeepEMD can be
> utilized as a fast surrogate for EMD wherever EMD is applicable, including tasks such as single-view
> point cloud generation or point cloud completion.
>
> W3. As discussed above, there is no inherent limitation in the DeepEMD architecture and we
> only choose to evaluate on point clouds as they represent uniform probability distribution and
> shape/object based point clouds datasets are easily available (number of samples and reasonable
> cardinality). We believe that the model could generalize well to these other datasets (LiDAR, etc.)
> but we would need it to scale to larger point clouds (or support sizes/ number of histogram bins).
> Further, test time generalization for a model trained on shapes might not be accurate enough
> for very different datasets but we can always train on those new datasets. The novelty in our
> method is to regress for the matching/coupling rather than the distance. We believe that this
> ground truth supervision with the matching allows the excellent generalisation capabilities to the
> model as compared to methods which regress for the distance itself. In particular, in the SetVAE
> application, the distribution of the generative model changes over iterations, hence, the population
> EMD also changes since one marginal (decoder output) is changing. We observe that the behaviour
> of DeepEMD is excellent during the whole process. This may be due in part to the noise we inject
> to the data samples during training of DeepEMD (explained in Sec. 4.1 and Sec. B.1). Note that
> we designed this data augmentation scheme before running any SetVAE related application, and
> did not revisit it, which makes us confident it is actually extremely robust.
>
> W4. The same limitations also arise with other methods like Hungarian, Sinkhorn, EMD, etc.
> We agree that investigating this line of work will be very useful. Concurrent research on attention
> models for increasing context length, reducing attention complexity, etc. can be applied to our
> method and bring in the gains seamlessly.
>
> Considering the strengths you recognized, the ’reject’ rating seems disproportionately severe. We
> are committed to enhancing the paper based on your feedback, so if you have any other concerns, we look
> forward to possibly addressing them during the discussion period. If this is not the case, we hope you
> will be open to changing your final rating, especially considering your acknowledgment of our paper’s
> contribution and potential usefulness.

---

### Official Review · Reviewer_sp92 · 2023-10-31

**Soundness:** 2 fair
**Presentation:** 3 good
**Contribution:** 2 fair
**Rating:** 5
**Confidence:** 3

**Summary:**

The paper introduces DeepEMD, a method for approximating the Earth Mover's Distance (EMD) with a significant speed improvement over traditional EMD calculation methods. The DeepEMD model is based on a multi-head multi-layer transformer, followed by a single-head full attention layer to predict the matching matrix. DeepEMD is evaluated on synthetic and real-world datasets, demonstrating its effectiveness in EMD approximation and gradient estimation. The model shows good generalization capabilities and can be used as a loss function in point cloud autoencoders. The conclusion also discusses potential future research directions, including faster transformer variants, architectural improvements, and extensions to other optimal transport problems.

**Strengths:**

-	The primary strength of this work is its focus on addressing the computational complexity of calculating the Earth Mover's Distance. By introducing DeepEMD, the authors significantly speed up the computation.
-	The results demonstrate that DeepEMD effectively approximates the true EMD, providing strong correlations between predicted and true distances. It also successfully estimates gradients, which is essential for various applications. This makes it a valuable tool for point cloud data analysis and potentially for other domains where EMD is utilized.

**Weaknesses:**

-	While the paper compares DeepEMD with other algorithms, it fails to compare against other state of the art works that compute EMD in terms of speed and accuracy.
-	The authors fail to make a tradeoff between speed and accuracy.
-	The authors talk about several works like CD, DPDist etc. in related works section which are faster than the proposed work. However, authors fail to compare their work with these works.

**Questions:**

-	How does this work compare against other recent works in terms of speed and accuracy?
-	Figure 8 shows DeepEMD to be performing 100x faster than Hungarian algorithm but has almost half the accuracy against the same algorithm. How much does DeepEMD sacrifice accuracy to achieve speed? Can you provide a more detailed speed vs accuracy tradeoff?
-	Is there a specific reason to use only 100 iterations for comparison? Can a better graph be provided that compares the methods over iterations?

---

> ### Author Response · Authors · 2023-11-16
>
> Thank you for your time reviewing our paper. We will try to address your concerns regarding
> related works, the weaknesses you highlighted, and your questions. We hope that it will lead you to reconsider your recommendation, and/or provide further insights for more improvements.
>
> W1. We experimented with several variants of Sinkhorn, including sinkhorn log, sinkhorn stabilized,
> greenkhorn, etc. However, we discovered that they either exhibited significantly slower performance
> compared to the vanilla variant, frequently failed to converge to a satisfactory optimal transport
> matrix within a finite timeframe, or sometimes exhibited both issues.
>
> W2. It is not clear what failing to make a tradeoff between speed and accuracy means here. Further, accuracy is a crude metric - when the model makes a slight assignment mistake for a point assignment, accuracy penalizes it while cosine similarity can still reveal the approximation error. The result on training auto-encoder with DeepEMD as a surrogate metric also emphasises the robustness of its approximation.
>
> W3. We do provide comparisons with CD and for the other related works we provide qualitative
> reasoning behind why DeepEMD is superior.
>
> Q1. We would be more than happy to provide this comparison, but there were no recent works on
> this task in our knowledge (other than the ones discussed above W1).
>
> Q2. Plese refer to the answer to W2.
>
> Q3. We can observe that Sinkhorn with 5 iterations is more expensive than DeepEMD and
> performs worse than DeepEMD. While further increasing the number of iteration might lead to a
> better convergence but the cost-performance trade-off in this regime is not very relaevant for our
> comparisons.

---

### Official Review · Reviewer_ijdr · 2023-11-01

**Soundness:** 2 fair
**Presentation:** 3 good
**Contribution:** 1 poor
**Rating:** 3
**Confidence:** 4

**Summary:**

The paper deals with the problem of computing (distance-based) similarity between 3D point clouds of shapes. The distance metric considered here is the EMD distance (earth mover distance) between 3D point clouds (considered as discrete distributions).
The authors have proposed a new approximate approach to compute EMD, which they refer to as DeepEMD. DeepEMD is significantly faster than classical methods (Hungarian algorithm) or the Sinkhorn method that allows an approximate solution to be computed.
The authors argue that their DeepEMD computation is efficiently enough that it can be used to compute EMD-based losses during training of other deep architectures. In particular they demonstrate promising results in training a point cloud VAE, by using DeepEMD instead of existing approaches to compute EMD.

**Strengths:**

The paper is well written and the general approach of using transformers to compare 3D point clouds and directly predict a matching matrix or the Earth movers distance (EMD) is a sound one. The architecture used by DeepEMD (the transformer-based model) produces accurate results at a fraction of the running time of some of the well-known existing methods.

**Weaknesses:**

The core idea of using transformers to compare pairs of 3D point clouds representing similar or overlapping shapes is not new. See [A, B]. This has been explored in the literature in papers such as these two papers. The transformer architectures in those papers also predict a matching matrix/scores that is then used to compute correspondences to solve a registration task. The proposed architecture is different from the ones in these papers but from what I can tell, the architectures/models in [A, B] appear to be superior as they allow the ability to deal with partial matching, outliers where the work proposed here cannot deal with such scenarios.

[A] RPM-Net: Robust Point Matching using Learned Features, Yew and Lee 2020.

[B] REGTR: End-to-end Point Cloud Correspondences with Transformers, Yew and Lee 2022.

The authors claim that their proposed approach to compute EMD approximately is significantly faster than existing techniques. They primarily consider two baselines, the well-known Hungarian method which computes EMD exactly and the well-known method that used entropic regularization to compute approximate EMD (Cuturi et al 2013) using matrix scaling / Sinkhorn Knopp algorithm. It is true that the proposed method is significantly faster than the Hungarian method and the Sinkhorn implementation tested here. However, there are a number of works in the literature which are either variants of the Sinkhorn method that have a better convergence than the original Sinkhorn method [C] or have investigated faster algorithms for computing EMD [D, E] that do not require any learning and therefore do not have the issue of overfitting/failure to generalize. A comparison with methods such as [C, D, E] is needed (perhaps there are a few other works in the literature) to make a stronger case for the proposed method.

[C] Near-linear time approximation algorithms for optimal transport via Sinkhorn iteration, Altschuler et al. 2017

[D] A Fast Proximal Point Method for Computing Exact Wasserstein Distance, Xie et al 2019

[E] Fast Sinkhorn I: An O(N) algorithm for the Wasserstein-1 metric, Liao et al 2022.

The title/abstract is not particularly appropriate. The proposed approach is aimed at computing EMD-based distance between 3D point clouds of geometric shapes. However, the title of the paper makes it sound like a generic method to compute EMD between arbitrary distributions has been proposed. Morever, the approximation factor in the proposed method is somewhat adhoc and not well analyzed, as some of the existing methods in the literature.

**Questions:**

Q1. Some of the implementation details should be clarified. How are the samples from the real datasets (ShapeNet, ModelNet40) used? What does "In order to improve and assess generalization, we augment the train and test splits with synthetic perturbations." (page 6) exactly refer to?

Q2. The sentences "These quite remarkable behavior .. as shown in Table 2." should be toned down a bit. What is the reason for this behavior? Further technical insights into what is being reported would make the claim stronger.

Q3. Why can't the linear-time EMD method of Shirdhonkar and Jacobs (2008) be used in practice. The sentence "However, their approach is limited to low dimensional histograms." needs to be explained further. Couldn't the 3D point clouds have been represented as coarser histograms and compared using their method?

Q4 (comment) the discussion of the method proposed by Amos et al. 2022 is not easy to follow. The distinction / similarity with their work needs to be explained more clearly.

---

> ### Author Response · Authors · 2023-11-16
>
> Many thanks for your thorough review of our paper. We address the raised concerns, weaknesses and questions below and hope that it will lead you to reconsider your recommendation, and/or provide further insights for more improvements.
>
> W1. RPM-Net [A] solves the point cloud registration task and uses an alignment algorithm
> (Sinkhorn iterations) to soft-align point features from feature extraction network (shared MLP).
> DeepEMD doesn’t solve the same task but instead can be utilized within RPM-Net for soft-
> alignment as an alternative to Sinkhorn iterations. Moreover, it does not have a transformer
> architecture.
> [B] also solves the point cloud registration task but does not use point matching/alignment.
> The input point clouds are converted into a smaller set of downsampled keypoints, and then
> contextualizes the key-point features through cross attention, and finally predicts the rigid trans-
> formation.
>
> Also DeepEMD is not necessarily limited to 3D point clouds as discussed below in more
> detail. The concern about outliers is also not applicable here as our goal is to approximate the
> optimal transport map and not point cloud registration. It is not clear what ’partial matching’
> implies here. If it is in regard with handling different cardinalities, DeepEMD can be easily adapted
> to this situation.
>
> W2. We experimented with several variants of Sinkhorn, including sinkhorn log, sinkhorn stabilized,
> greenkhorn (ref [C], POT provides an implmentation), etc. However, we discovered that they ei-
> ther exhibited significantly slower performance compared to the vanilla variant, frequently failed
> to converge to a satisfactory optimal transport matrix within a finite timeframe, or sometimes
> exhibited both issues. Further [D] has a similar complexity as Sinkhorn. Also note that [D] demonstrates that ”regularized
> variations with large regularization parameter will degrade the performance in several important
> machine learning applications, and small regularization parameter will fail due to numerical sta-
> bility issues with existing algorithms.” which applies to our case for using Sinkhorn and variants
> for benchmarking. The methods in [E] is limited to the Wasserstein-1 metric and doesn’t generalize for Wasserstein-p.
>
> W3. Although we choose to evaluate on 3D point clouds, note that DeepEMD does not assume
> anything about the input data and does not process 3D point clouds in any special way but works
> directly with the raw features. Thus the limitation to 3D objects is not significant.The main goal
> is to approximate EMD between two distributions which is ’similar’ to point cloud matching when
> working withpoint clouds as uniform distributions. In principle, DeepEMD can be utilized as a
> fast surrogatefor EMD wherever EMD is applicable.
>
> Q1. We have provided more details on augmentation in Appendix Section B.1
>
> Q2. An intuitive explanation for such behaviour could be : The prediction of matching requires
> to have reasoning able to match pieces of structures that may be distant, and taking into account
> what has already been matched. It is unclear how we could even shape the data to use convnets,
> for example. Transformers are extremely relevant here because we are dealing with sets without
> any ordering prior information, and we do not know in advance the structural complexity of the
> distributions (e.g. number of ”pieces”, geometric degrees of freedom) and an attention model does
> not summarize the said structure into a fixed dimension embedding.
>
> Q3. The paper mentions in the abstract ”We present a novel linear time algorithm for approx-
> imating the EMD for low dimensional histograms...”. They provide a discussion in Sec 4.1.4 and
> Sec 4.2.
>
> Q4. Amos et al. 2022 presents an amortization based method and the overall approach works in
> two stages : 1.) A Meta-OT model (neural network) is trained to predict approximate OT solution.
> 2.) The approximate solution is then used to initialize Sinkhorn’s algorithm to further refine and
> output the final solution. As such it is orthogonal to our work as DeepEMD can be used in stage
> 1 for predicting the approximate OT solution. Note that they consider the entropic regularization
> problem and do not approximate EMD itself. We will be happy to include this in the paper.

---

> > ### Comment · Reviewer_ijdr · 2023-11-20
> > **Thanks, some follow-up questions**
> >
> > I appreciate your detailed response and clarifications. However, could you clarify your responses to Q3 and Q4 and clearly explain what benefit your proposed method has over those two existing approaches. Please clarify why these methods could not be included in the experimental evaluation? Or is there evidence elsewhere (i.e. in another paper) that shows that such baselines would be inferior to the baselines included in the paper. In that case, please share a reference we can check.
> >
> > One more follow-up question. Regarding the argument you presented in paragraph (W2), are the observations regarding the Sinkhorn variants and (D,E) reported somewhere in the paper? For the task at hand, does it matter that (E) is limited to Wasserstein-1? Is the generalization to Wasserstein-p important?
> >
> > Regarding my earlier comment about 'partial matching', it has to do with the fact that in point-cloud registration it is common situation that 3D points in one point cloud scan are absent (or not visible) in the other scan and vice versa. See the RPM-Net paper figures and description for more details.  Can your method be used to compute the EMD for such data (with partial overlap)?

---

> > > ### Author Response · Authors · 2023-11-22
> > >
> > > Thank you for your response.
> > >
> > > The linear-time EMD method of Shirdhonkar and Jacobs is applicable to histograms. Since we are working with point clouds as distributions, it doesn't directly apply to our setup. Further if we still convert the point clouds to histograms, it will likely lead to very sparse histograms and gradient computation will not be trivial which is needed to use it as a loss. Moreover, we could not find the implementation for their method.
> > >
> > > Regarding MetaOT, we believe its not a direct competition to our method. As discussed before, when working with point clouds, we can use DeepEMD as the meta model in their framework. Note that they do not discuss any specific architecture for computing the matching, which is our main contribution.
> > >
> > > The observations relating to W2 are currently not reported in the paper, but indeed they are very relevant and we will add it to the final version of the paper. The choice of the underlying metric (L1 for Wasserstein-1 and Lp for Wasserstein-p) leads to very different solutions for the matching, thus depending on the domain it can be very important. Since DeepEMD is a learning-based method, it is agnostic to the underlying metric, and it only sees the metric through the ground truth matching matrix.
> > >
> > > Indeed, DeepEMD is applicable as a surrogate for EMD wherever EMD is applicable. For this specific use-case, we would need to extend DeepEMD to work with different cardinality of point clouds in the pair, and thus allow for mass splitting. We do plan to consider this setting in future (see Sec. 5).

---

### Official Review · Reviewer_Fn5r · 2023-11-01

**Soundness:** 3 good
**Presentation:** 2 fair
**Contribution:** 2 fair
**Rating:** 5
**Confidence:** 4

**Summary:**

This paper presents a deep learning based approach DeepEMD for estimating EMD from two point clouds. DeepEMD is much faster then the O(N^3) EMD, and also achieves better performances compared with EMD, CD or Sinkhorn.

**Strengths:**

1. The proposed EMD is much more efficient than EMD or Sinkhorn.
2. Fig. 5 shows that the estimated distance from DeepEMD do not have large errors compared with ground truth EMD.

**Weaknesses:**

1. The evaluation can be improved. All the experiments are conducted at object level. However, the authors do not evaluate DeepEMD under scenes or other types of point clouds. I thinks DeepEMD is limited to the 3D objects since it is only trained with 1024 points under only object-level datasets. A discussion of generality of DeepEMD is needed.
2. More 3D visualization can improve the representation. The paper do not contain any 3D visualization for reconstruction, generation, etc. This makes the paper lack intuitive qualitative comparison.
3. It will be much more convincing if the authors adopt DeepEMD as a loss to previous generative models (e.g. single-view point cloud generation, point cloud completion) and report the performance compared to the baseline models.

**Questions:**

1. I think that the argmax in Eq.(10) is not differentiable, how did you solve that?
2. Will DeepEMD still be efficient with a large point number?
3. How long dose it take to train DeepEMD, and how much GPU do you use?

---

> ### Author Response · Authors · 2023-11-16
>
> We thank the reviewer for the their valuable feedback and time. We address the main concerns raised below. We hope that it will lead you to reconsider your recommendation, and/or provide further insights for more improvements.
>
> W1. Although we choose to evaluate on 3D point clouds, note that DeepEMD does not assume anything about the input data and does not process 3D point clouds in any special way but works directly with the raw features. Hence, there is no limitation per se of our method to 3D objects.
>
> W2. This is indeed a very good point, such representation will be added to the final version.
>
> W3. We do show that DeepEMD (see Sec 4.4) as a loss works well for training an  auto-encoder as well as a variational auto-encoder. In principle, DeepEMD can be utilized as a fast surrogate for EMD wherever EMD is applicable, including tasks such as single-view point cloud generation or point cloud completion. In addition to good approximation to the EMD and its gradient, we show the most straight-forward evaluation by training  auto-encoders which are completely based on the distance metric and thus appropriate in assessing the quality of DeepEMD as a surrogate metric.
>
> Q1. We only use eq. 10 during inference and thus it doesn't need to be differentiable.
>
> Q2. We do show in Sec 4.3 the test time performance with point clouds as large as 8196 and observe that it performs excellently. With transformer architectures which are either faster or scale up to larger context sizes, we should be able to scale up DeepEMD.
>
> Q3. In general, we typically need to train a single DeepEMD model for a given dataset (note that this step may not be needed given the generalization capabilities of the model). Further, training DeepEMD takes about 4 mins for an epoch of training on a single NVidia RTX 3090 and 16 GB memory with a batch size of 16 (we train for 75 epochs). During inference, it needs 0.8 GB GPU memory. Also note that the inference and training times can be improved by 5-8x with efficient attention implementations, eg. flash attention. When used as a loss, the parameters of DeepEMD are fixed and gradients for the same are not needed, thus its memory footprint is very limited.
>
> We hope this response has clarified some of your questions, particularly concerning why mesh-based approaches are not suitable candidates for baselines in this case. Please let us know if anything remains unclear, as we would be happy to provide further information if necessary.

---

### Meta-Review · Area_Chair_knk3 · 2023-12-06

**Metareview:**

The paper's primary claim of proposing a faster and more efficient method to compute Earth Mover's Distance (EMD) and its application to point-cloud registration task is ambitious. However, several reviewers have rightly pointed out that a more comprehensive evaluation of DeepEMD against existing alternatives is essential to substantiate these claims. Although the authors have addressed some of the issues raised in the rebuttal, they fall short in providing experimental evidence to support their arguments about the limitations of current approaches.  Similarly, concerns regarding the application to point-cloud registration—a well-explored area—have not been adequately addressed. The paper would benefit significantly from a more detailed comparison with existing methods in this domain. In light of the concerns raised during the review process, I recommend the rejection of this paper.

**Justification For Why Not Higher Score:**

Not good enough validation.

**Justification For Why Not Lower Score:**

N/A

---

### Decision · Program_Chairs · 2024-01-16

Reject